# Food Safety Trust, Risk Perception, and Consumers’ Response to Company Trust Repair Actions in Food Recall Crises

**DOI:** 10.3390/ijerph17041270

**Published:** 2020-02-17

**Authors:** Chuanhui Liao, Yu Luo, Weiwei Zhu

**Affiliations:** 1School of Economics and Management, Southwest University of Science and Technology, Mianyang 621010, China; liaochuanhui@163.com (C.L.); luoyu9602@163.com (Y.L.); 2School of Marxism, Hefei University of Technology, Hefei 230009, China

**Keywords:** consumer trust repair actions, risk perception, coping-appraisal perception, self-sanction, information-sharing, protective behavioral intention

## Abstract

Food recalls have severe impacts on the operation, reputation, and even the survival of a recalling company involved in a crisis, with consumer trust violation being the immediate threat to the recalling firm. The involved firms adopt trust repair strategies and release messages relevant to these actions to the public. In this research, we developed a conceptual model to analyze consumers’ general responses to the food recall, and we then compared the effect of two types of consumer trust repair strategies, i.e., self-sanction and information-sharing. The results show that consumer food safety trust has negative impacts on consumers’ protective behavioral intention during a food recall crisis. In the scientific-evidence sharing group, consumers have a higher risk perception, coping appraisal efficacy, information-seeking tendency, and protection behavioral intention. However, consumers’ food safety trust fails to predict protection behavioral intention because scientific-evidence actions can either be regarded as an explanation and self-serving, or as useful facts and solutions. Self-sanction actions overcome the disadvantages of information-sharing actions, but consumers still require information on facts of and solutions to the crisis. Therefore, it is recommended that recalling firms combine these two strategies in the case of consumer trust repair in food recall crises. Furthermore, the involved firms are encouraged to employ a third party to release the scientific evidence.

## 1. Introduction

Food recalls have severe impacts on the operation, reputation, and even the survival of a recalling company involved in a crisis [1], but one of the most critical impacts of a food recall is the violation of consumer trust [2]. As a valuable resource, trust is very important in the consumer–organization relationship [3,4,5]. However, it is fragile and vulnerable to threats [2]. A food safety crisis can be regarded as a type of consumer trust violation [6], which exerts negative effects on the consumers’ cognitions and emotions toward the company [7], increasing the consumers’ estimation of the efficacy of the suggested protective responses [8], leading to a reduction of the consumers’ active action [3], activating great consumption change [9], and even destroying behavior [6]. During the response process, consumers might be exposed to large amounts of information released from different channels by varieties of parties, resulting in an overestimation of the probability and vulnerability of potential damage [10]. Moreover, this will influence consumers’ response behavioral intention [11,12]. However, in previous studies, few researchers have taken the process mentioned above as a whole. Therefore, we set our first research objective as exploring consumers’ crisis-coping processes and response behaviors during food recall crises.

Information communication is a crucial factor influencing seller-buyer trust. In a product-harm crisis, the organizations involved usually take actions and release pre-packaged information subsidies and well-prepared stories about firm activities, to influence consumer negative perception about the firms [13]. Hence, in a food recall, the involved company will take trust repair actions and then release the concerned information through intermediaries, in order to simplify consumers’ information seeking and influence consumers’ perceptions of the affair and their behaviors. Previous studies have focused on routine strategies, with less attention having been paid to hostage-posting [14] and financial reimbursements [2,15,16]. Hostage-posting refers to a commitment device employed to reduce uncertainty and resolve social conflict, while self-sanction is a type of voluntary hostage-posting used in an uncertain situation [16]. In the context of Chinese spirit recall, the recalling manufacturer can recall beyond the legal liability, for example, by recalling more than just the identified lot, or provide more remedies to the customers. Therefore, according to Nakayachi and Watabe [16], we set the second research objective as exploring how repair strategies (self-sanctioning and information sharing) affect consumers’ response behaviors differently. 

To address these two research objects mentioned above, we developed a conceptual model to assess consumers’ general risk perception and response behaviors. In addition, we compared the effective of two different trust repair strategies, namely the self-sanction strategy and the information-sharing strategy, in shaping consumers’ reaction behavior. This research was conducted by formulating questionnaires mirroring actual Chinese spirit recalls that have occurred in China. In March 2016 (CFDA Notification of unqualified Chinese spirits in the quality-inspection of the 42nd batch (No 59, 2016), available on http://samr.cfda.gov.cn/WS01/CL1687/147481.html (accessed at 12 October 2016)), the China Food and Drug Administration (CFDA) announced that 42 lot samples were found to be unqualified in Chinese spirit quality inspections during October and December of 2015. In these 42 lots, 5 identified as containing cyanide and 10 as being associated with an abuse of food additives were recognized as risky food and recalled. These Chinese spirit safety scandals, together with the former plasticizer storm and Chinese spirit-blending affairs, have great impacts on the development of the Chinese spirits industry. Therefore, in this research, we mirrored these actual Chinese spirit recalls and developed fictional recall announcements, with any dates and identifying information altered, to analyze consumers’ reaction to different trust repair strategies.

This article is organized as follows. After the introduction in section one, the theoretical framework and hypotheses are presented in section two. In section three, the research method is presented. The results and discussion are elaborated upon in section four and five, respectively. Then, the conclusions, suggestions, and limitations are presented in the last part.

## 2. Theoretical Framework

Protection Motivation Theory (PMT) is used to explain people’s behavioral change in response to a health risk, an environmental risk, and food safety hazards [17,18,19]. As a comprehensive persuasion model, PMT successively analyzes the effects of two cognitive processes (the threat appraisal and the coping appraisal) over the motivation and intention of protective actions. People first evaluate the gravity (severity) of, and exposure (vulnerability) to, a risk, which is defined as an estimation of threat appraisal [17,20,21]. They further evaluate the coping appraisals, with response efficacy, to estimate the efficacy of the suggested solutions in order to remove the threat, and the self-efficacy in order to assess their perceived belief and confidence in their ability to handle the risky issue [22]. The results of these two estimations further activate the protection motivation and behavioral intention [17,19,20]. 

The Heuristic-systematic model (HSM, [23,24] assumes that the formation of individuals’ attitude is based on information processing [25]. It indicates that information seeking is the antecedent factor of behavioral change intention. People who incline to seek information will have a significant tendency to conduct protective behavior to reduce the harm [23]. Though the PMT model has been widely used in risk situations [17,18,19], it has not been used to analyze consumers’ behavioral reactions to food product harm hazards. Since the main effect of a food product hazard lies in its threat to consumers’ health, it is appropriate to use it in this situation. Moreover, information seeking and processing have not been considered in the PMT, despite the fact that they can affect people’s risk perception and coping appraisal efficacy (CAE). It has been indicated that food safety is generally regarded as credence, which cannot be observed by consumers [26]. Hence, food safety information can be regarded as risk information aiming to reduce the consumer uncertainty while making purchasing decisions [26]. Previous studies have certified the importance and effectiveness of applying HSM in risk situations [27]. Therefore, we can incorporate PMT and HSM to develop a more comprehensive model for exploring consumers’ reactions to the food recall context.

Furthermore, trust acts as an essential antecedent factor of risk perception and coping perception in crisis situations [8,12,28,29,30]. In an industrialized society, consumers rely more on social trust to deal with emergent and risky affairs that require large amounts of knowledge and skills [31]. Hence, we set consumer food safety trust as the antecedent of the psychological process in the context of food recalls [8].

Based on the discussion above, we developed the theoretical framework shown in Figure 1. In this model, in food recall situations, consumers first evaluate the severity and vulnerability of the risk (risk perception), and then estimate the efficacy of the proposed coping measures and their ability to conduct these protective behaviors, known as the coping appraisal efficacy (CAE). During this process, consumers may experience information inefficiency and are inclined to seek extra information [8]. Lastly, consumers make their consumption decisions based on these psychological, cognitive, and information seeking processes.

### 2.1. Consumer Food Safety Trust 

Trust is of great importance in the consumer–manufacturer relationship [8,29]. With the advancement of society, economy, and technology, people cannot possess knowledge sufficient for all risks and threats. They have to resort to social trust and rely on governments, manufacturers, and experts to make response decisions in times of product harm crisis [31]. This is the case in the area of food safety since large amounts of advanced technologies have been adopted in food production, such as genetically-modified food, food additives, and functional foods [19]. In the domain of food safety, the trustees include food supervision departments, consumer associations, experts, and food manufacturers and retailers [32]. The more consumers rely on these trustees, the lower the risk level they perceive [32]. At the same time, a consumer’s food safety trust may influence their CAE. When they have a high trust level toward food safety, they tend to accept the responses provided by the trustees [8,28,29]. Hence, we proposed the following hypotheses:

**H1/2/3**:
*Consumer food safety trust has a negative effect on risk perception (H1), CAE (H2), and protective behavioral intention (H3).*


Trust is one of the prerequisites for effective risk communication [8,27,33], and Liao (2013) [34] indicates that risk information changes the perceived risk and coping efficacy that consumers feel, inducing uncertainty. The increase in uncertainty motivates people to seek more information and help to analyze the hazards so that they can react to a certain risk. Therefore, when the trusted party provides the risk information, consumers who have a higher trust and lower uncertainty about the accuracy and usefulness of the information do not feel information insufficiency and thus decrease their information search behavior. Hence, we formulated hypothesis H4:

**H4**:
*Consumer food safety trust has a negative effect on risk information seeking intention.*


### 2.2. Risk Perception and Coping Appraisal

In PMT, in a risky situation, people conduct threat appraisal and coping appraisal analyses to make behavioral decisions. Threat appraisal includes assessments of risk severity and vulnerability, and these two evaluations are usually integrated into risk perception [34]. A coping appraisal includes response efficacy and self-efficacy [17]. When people feel a high-perceived threat, they are more cautious and actively take protective behavior to reduce the harm [8,35]. Chen [19] also confirmed that the food safety risk perception held by consumers in Taiwan increases their willingness to buy safe food and induces higher protective behavioral intentions. Generally, when consumers believe that the protective measures provided by the trustees are effective, they will choose and adopt the recommended protective measures. Therefore, we hypothesize that 

**H5/H6**:
*Risk perception (H5) and CAE (H6) have a positive influence on the protection behavioral intention.*


In the RISP (Risk Information Seeking and Processing) model, the perceived information ‘gap’ is the main motivation of the information search [27]. Changes in risk perception and CAE will evoke affection responses (usually fear), motivate information demand, and further induce actual information seeking behavior. Shan [36], Zhang [37]), and Terpstra [38] have confirmed that people in risky situations demonstrate more information seeking intentions and behavior. Hovick [11] and Wei [39] demonstrated the indirect and direct effect of risk perception on the information process. At the same time, when consumers feel high CAE, they think highly of the effects of the recommended response measures [20]. In this situation, consumers with more food safety trust may accept the protective measures provided by experts and companies [23], and no longer seek more information. The study of Verbeke [8] also indicates that in a food safety crisis, the Dutch consumers’ risk perception significantly and positively influences information seeking intentions, while the effect of coping appraisal is not significant. Therefore, we propose the following hypotheses:

**H7**:
*Risk perception positively affects information seeking.*


**H8**:
*Coping appraisal has a negative influence on information seeking.*


### 2.3. Protection Behavioral Intention

In marketing, behavioral intention is an indicator that demonstrates the consumers’ decision to stay with or defect from the company [40]. It has been used to predict consumer responses to product-harm crises [9,10,41]. In terms of food crises, Li [41] investigated the consumers’ willingness to pay (WTP) after the 2010 egg recall and found that the WTP for organic eggs significantly increased with negative information of recall. Additionally, balanced information (positive and negative) will mitigate the increase. Peake [9] also indicates that media reliance has strong and direct effects on broad consumption changes in the context of fictional food recalls. Furthermore, HSM demonstrates that information seeking is the antecedent factor of behavioral change intention. Individuals who tend to seek information are inclined to conduct protective behavior to avoid the harm [23]. Therefore, we developed the following hypothesis:

**H9**:
*Information seeking positively affects protective behavioral intention.*


### 2.4. Crisis Communication and Trust Repair Messages

Information communication is a crucial factor influencing seller-buyer trust. When examining the role of crisis communication in product crisis situations, the theory of consumer trust repair and its various iterations have become the most dominant focus of research since its start in the mid-2000s [2,16,42]. It has been applied in various domains, such as various manufacturers [16,43], service providers [43,44], and E-commerce businesses [45]. However, only two papers have focused on consumer trust repair in food crises. Giraud [46] found that a private label system had a positive effect on consumer trust restoration after the mad cow crisis in Europe. Cleeren [47] also indicate that brand advertising was more effective in trust repair for stronger brands than weaker brands in the context of the food harm crisis of peanut butter in Australia. 

In this paper, we choose self-sanction (one type of hostage posting) and information-sharing strategies as the targeted strategies and explore their effect on consumers’ responses based on the following reasons: (1) Of all the consumer trust repair strategies, hostage posting is less studied [48]. Only two articles have explored the effectiveness of monitoring and self-sanction. Dirks [14] indicate that two ‘substantive’ responses, i.e., penance and regulation, can increase organizational trust. Nakayachi and Watabe [16] also found that voluntary hostage posting raised participants’ organizational trust. However, these two studies deal with organizational trust, and we know less about the effectiveness of a self-sanction strategy on consumer trust repair in the context of food recall hazards; (2) information sharing is an important action in trust repair [2,15]. The explosive diffusion of information caused by information dissemination, especially negative news, poses new challenges for manufacturers when attempting to repair and restore consumer trust. Current studies demonstrate that information communication is an important way of eliminating mutual suspicion coordinating expectations and ultimately fostering trust [49,50]. While most of the information released is news denoting ‘good doings’ and the ‘responsibility’ of the involved companies, few studies have explored the effectiveness of releasing the ‘scientific evidence’. To the best of our knowledge, only two papers have discussed this issue. Wen et al. [51] have discussed the effectiveness of ‘scientific evidence’ provision in image repair for American beef in Taiwan. The result indicates that although the US government used scientific data as support, the rhetoric was not successful in the end. Li [41] demonstrates that providing mitigating information was effective in alleviating the decrease of ordinary egg purchases after the 2010 egg recall. Meanwhile, we know less about how these two strategies influence consumer trust repair and how consumers respond to this information. Hence, in this study, we adopted an empirical examination and proposed the following research question:

RQ: How do consumer responses differ depending on the two types of trust repair messages (self-sanction and information sharing) in food recall crises?

## 3. Research Method

### 3.1. Questionnaire and Measures

A questionnaire survey was used to collect data. There were four parts in each questionnaire. Part 1 presented the aim and expressed the gratitude for the participants. In part 2, we provided the simulated scenarios. Then, in part 3, items concerning the constructs were presented, followed by demographic data collection in part 4. According to the theory of trust repair and the study aim, we set the information of scientific evidence and self-sanction strategies as the backgrounds of scenario A and B, respectively. The scientific evidence for food additives disclosed in scenario A was as follows: 

(1) Cyclamate is a synthetic sweetener, which is widely used in beverages and pastries, according to the national standards, but not in Chinese spirits; (2) the unqualified samples may be caused by the improper use of raw materials and accessories; (3) according to the ADI value set by the Joint FAO & WHO Expert Committee on Food Additives (JECFA), an adult of 60 kg has to consume 9.9 kg of Chinese spirits a day for it to be harmful.

Scenario B provided the background of the self-sanction measure, described as providing more remedies beyond the legal liabilities. Scenario B was as follows: 

The Chinese spirits of X brand produced on 8 March 2014 (lot number 20140308), and on 18 June 2015 (lot number 20150618), were found to be unqualified, due to the detection of cyclamate by the CFDA during a sampling inspection. To be a responsible company, this company decided to recall all the products produced in 2014 and 2015, in addition to the detected unqualified lots. Retailers and consumers who hold these products should please contact our product recall office as soon as possible. We are sorry for the inconvenience. 

Based on previous literature, we used multiple items to measure the constructs, with slight modifications, in order to conform to the food recall context. A seven-point Likert scale was used, with 1 for strongly disagree and 7 for strongly agree. All the constructs, items, and references are shown in Table 1.

### 3.2. Sample and Data Collection

After a pre-test, the formal questionnaire survey was conducted through online and offline channels. The questionnaire was posted on Wenjuanxing, a popular electronic data collection platform, from 20 December 2016 to 28 February 2017. At the same time, trained surveyors distributed the questionnaire to consumers in supermarkets, kiosks, and stores during the winter vacation time. All the participants anonymously answered the questions and were assured that their responses would remain confidential. Copies with missing data and identical responses in most of the items were deleted. Finally, there were 461 usable questionnaires, with 158 on-site surveys and 303 online surveys. Of all the participants, there were 290 females. Fifty-six percent of the respondents were 36–50 years old, 74.41% were married, and 52.71% had completed senior high school education and above. More than half of the respondents earned less than ¥4000 each month. We used a χ^2^ test to evaluate whether the sample represented the general distribution of the Chinese population. The results confirmed that the characteristics of gender and age in the sample were representative of the population. Generally, the sample can represent the population to some extent. 

Since data were collected from two channels, i.e., online and offline channels, at the same time, common method bias (CMB) may exist. Harman’s one-factor test was employed, and the results indicated that all the items were divided into five constructs with the criterion of eigenvalues greater than 1. All these constructs explained 73.03% of the variance, with the first one representing 28.02%. Hence, the data was fit for further analysis.

## 4. Results

### 4.1. Reliability and Validity Assessment

We employed exploratory factor analysis (EFA) and confirmatory factor analysis (CFA) to evaluate the construct reliability. As shown in Table 2, all the values of Cronbach’s a and composite reliability are higher than 0.8, and the proportion of variance explained (R2) is between 0.42 and 0.892, indicating an acceptable reliability. Secondly, content, convergent, and discriminant validities were tested. Since all the items were adopted from previous studies, with minor revision made under the research situation, content validity is supported. Convergent validity was assessed through factor loadings, composite validity, and average variance extracted (AVE). All the factor loadings and composite reliability values are near or greater than the benchmark of 0.7, and the AVEs range from 0.63 to 0.83 (Table 3), supporting a good convergent validity. As for the discriminant validity test, the square roots of the AVEs are higher than the correlations between each pair of constructs (Table 4). Therefore, the data exhibited a good liability and validity and was ready for further analysis. CFA was employed to test the measurement model. The results show that the indicators (χ^2^/df = 2.22, TLI = 0.932, NFI = 0.913, CFI = 0.945, RMSEA = 0.038, and SRMR = 0.061) conformed to the recommended thresholds, supporting the overall fitness of the CFA model.

### 4.2. Structural Model Describing Consumers’ Responses

A structural equation was used to estimate the proposed model. The results show that all the indicators (χ^2^/df = 1.572, TLI = 0.981; NFI = 0.965; CFI = 0.987; RMSEA = 0.035; SRMR = 0.049) indicate an acceptable model fitness. Figure 2 demonstrates the path coefficients (β), significance level (*p*), and explained variance (R2). The results of the SEM analysis indicated that food safety trust, risk perception, and coping appraisal efficacy explained 33% of the variance in information seeking, which subsequently explained 55% of the variance in protection behavioral intention, indicating a medium to high interpretation power. However, food safety trust explained 16% of the covariance of coping appraisal efficacy, indicating a small interpretation power. In general, the results of the SEM analysis indicated a satisfactory fit [19]. 

As shown in Figure 2, the consumer food safety trust is negatively correlated with risk perception (β = −0.405, t = −7.018), CAE (β = −0.574, t = −11.722), information seeking (β = −0.236, t = −3.905), and protective behavioral intention (β = −0.151, t = −2.846), thus supporting hypotheses H1, H2, H4, and H3. As an important mediating variable in PMT, the coping perception mediating process shows a significant effect on food safety trust and protection behavioral intention. On the one hand, a higher risk perception leads to a higher protection behavioral intention (β = 0.427, t = 7.128) and more information-seeking (β = 0.363, t = 5.957), supporting the hypotheses of H5 and H7. On the other hand, CAE has a positive effect on behavioral protection intention (β = 0.12, t = 2.932), supporting H6. Finally, information seeking has a positive and significant effect on protection behavioral intention (β = 0.274, t = 4.876). Therefore, H9 was supported. 

The significant effect of CAE on information seeking is consistent with the proposed assumption, but with the opposite direction. Therefore, H8 is not supported. To further analyze the role of CAE, we conducted a quadratic regression. We set CAE and the square of CAE as independent variables, and information seeking as a dependent variable. 

The results (Table 4) show that CAE positively affects information seeking, while the square of CAE negatively influences information seeking. Hence, the relationship between CAE and information seeking demonstrates an inverted U-shape. This implies that when a consumer holds a low level of CAE, they have perceived information insufficiency, leading to more information seeking. 

### 4.3. Results of Trust Repair Strategy Comparisons

We evaluated the contrasts between the two trust repair strategy groups, i.e., the information-sharing and self-sanction groups. We first employed an independent sample test and confirmed significant differences between these two groups (Table 5). We also employed a single multi-group analysis to analyze these differences. The structural model for each trust repair strategy group is demonstrated in Figure 3. All indicators conform to the recommended benchmark and show that the model and data fit well (χ^2^ = 694.51, df = 300, χ^2^/df = 2.315, TLI = 0.918, NFI = 0.901, CFI = 0.935, RMSEA = 0.054, and RMR = 0.078). Moreover, the explanatory powers of the three data sets are almost the same, indicating the robustness of the model. In Figure 3, we only demonstrate the significant paths to emphasize a visual comparison of the differences between these two groups.

Differences between the full model and the two groups are presented in Figure 2 and Figure 3. There are two differences between the full model and the information-sharing group. First, food safety trust has a significant effect on the protection behavioral intention in the full model, but not in the information-sharing group. Second, CAE positively influences information-seeking in the full model, but fails to do so in the information-sharing group. There is only one difference between the self-sanction group and the full model group. In the self-sanction group, information-seeking is no longer an antecedent factor in predicting protection behavioral intention, while it is in the full model.

There are differences between these two groups. In the self-sanction group, food safety trust negatively influences protection behavioral intention and CAE positively influences information-seeking. However, these two paths failed in the information-sharing group. Moreover, in the information-sharing group, information seeking activates higher protection behavioral intention, but fails to do so in the self-sanction group. The research certifies the differences in the effect of the two kinds of trust repair strategies.

## 5. Discussion

This study has explored how consumers respond to food recall crises and examined the effects of two types of consumer trust repair strategies. The main results and explanations will be provided below.

### 5.1. Antecedents of Protective Behavioral Intention

As for the first research objective, most of the findings are in accordance with our hypotheses and previous research. Consumers holding higher levels of food safety trust usually have a lower risk perception and CAE. They feel less information inefficiency, exhibit less information-seeking behavior, and are less likely to conduct protective behaviors. In this paper, consumers’ food safety trust was measured by consumers’ trust in the CFDA and its subordinates, food manufacturers and retailers, and third parties [32]. Hence, consumers with a lower level of food safety trust may not rely on these stakeholders and take shortcuts by adopting information and response suggestions provided by these stakeholders, and may conduct response behaviors accordingly, thus increasing the information need and information seeking intention [29]. Moreover, consumers with a lower level of food safety trust may feel a higher level of coping efficacy. They may reject the suggested coping behavior provided by experts and CFDAs because most of the suggestions are simple and superficial efforts to deal with food safety harm, such as checking, dropping, and returning the recalled food. These measures are not sufficient for solving food safety problems. Consumers with a low level of food safety trust tend to spend time and effort seeking information about the abuse of food additives because they are concerned about their health.

Risk perception and CAE are predicted to positively influence both information seeking and protective behavioral intention, conforming to our hypotheses and previous studies [8,19]. When faced with a food crisis, consumers have a higher risk perception because of information asymmetry, which leads to more fear and uncertainty of the potential harm [52]. Besides, a higher uncertainty and coping appraisal may facilitate their protective behavioral intention [8,19]. Moreover, risk perception and coping appraisal have a significant positive effect on information seeking, which conforms to previous studies [27,36]. However, coping appraisal positively influences information seeking, with the opposite direction of the coefficient compared to our hypothesis. A further estimation using a quadratic regression shows that the relationship between CAE and information seeking is an adverse U-shape. The probable reason for this may be explained as follows. Before the turn-point, consumer food safety trust is at a low level, and people have less knowledge of food additives, so consumers may perceive a low competency andself-efficacy in solving the problem [33,53]. Therefore, they may seek further information, leading to a positive relationship between CAE and informational seeking intention. With the accumulation of information, knowledge, and experience, consumers’ self-efficacy evaluation will gradually increase. As the consumer’s CAE exceeds the turning point, consumers are competent at assessing the effectiveness of recommended measures and other available measures. Therefore, coping CAE in the second half demonstrates a negative correlation with information seeking. Generally, the positive impact of risk perception and CAE of information seeking highlight the critical role of information searching and communication in consumer reactions to food harm crises. 

### 5.2. Trust Repair Action Messages and Consumers’ Reaction Behavior

The second aim of this research was to compare the effectiveness of two kinds of trust repair action information in the context of food recalls. The results demonstrated that there were three main differences between these two groups. First, in the self-sanction group, food safety trust has negative impacts on consumer’s protective behavioral intention. However, it is insignificant in the information-sharing group. A reasonable explanation for this insignificant effect may be as follows. On the one hand, although consumers hold a negative perception of the recall, they also attach importance to the scientific information provided by the manufacturers [41], resulting in a reduction of the consumer’s intention to take protective actions, to a certain extent. On the other hand, some consumers would argue that providing scientific-evidence information is an act of explanation and self-serving [44,51], especially with the comparative information about the usage and dosage of other types of food additives. This result conforms to the study of Wen et al. [51]. To repair the image of American beef in the “mad cow” time in 2010, the US government provided statements of scientific research; however, the US officials and scientific data were not perceived as objective sources, and defenses of quality of US beef were perceived as self-serving [51]. Providing scientific data is regarded as an apology with an explanation or denial with an explanation [44], which will lead to a lower trust repair efficacy and higher protective behavioral intention. In comparison, in the self-sanction group, the recalling firm provides financial compensation and recall beyond the legal liability (recalls more products than required). This behavior information signals that the recalling firms are actively implementing social responsibilities, and thus helping to repair and restore consumer trust and decrease broad consumption changes [16]. Therefore, when provided with scientific-evidence information, the impact of food safety trust on behavioral intention is not significant.

Second, consumers’ coping appraisal in the self-sanction action group positively influences information seeking, but is insignificant in the information-sharing action group. In the self-sanction action group, when the recalling firm provides financial compensation and recalls beyond the legal reliability, they are conveying a good image of social responsibility, indicating the firm’s promise that this wrongdoing will not happen in the future. However, consumers still need to search for more information to make decisions because of a perceived lack of technical solutions. Conversely, in the information-sharing action group, the effect of coping appraisal on information-seeking intention is insignificant. The possible reason for this may be as follows. On the one hand, providing scientific evidence during the product recall crisis can be regarded as self-serving or as an apology with an explanation, which decreases the consumer food safety trust and enhances their evaluation of the efficacy of consumption change responses [51,54]. Hence, consumers may feel that it is unnecessary to seek more information. On the other hand, the extensive use of additives in foods disclosed in the scientific information may make consumers confused and anxious about the severity and ubiquity of food additive problems. They may thus perceive less confidence in the efficacy of the recommended responses and hold a low response efficacy. At the same time, understanding additives requires knowledge of chemistry, biology, and food science which is beyond most of the consumers [55]. Consumers are incompetent at finding, judging, and getting rid of the additives in the food, leading to a low level of self-efficacy [36]. Therefore, people perceive the information gap and will seek more information concerning food additives to make decisions.

Lastly, in the information-sharing action group, the relationship between information searching and protective behavioral intention was not significant, whereas it was significantly positive in the information-sharing action group. As mentioned before, financial compensation and recall beyond the legal liability will increase consumers’ recognition of companies’ social responsibility performance and consumer trust in enterprises and products. Therefore, consumers will tend to maintain their pro-corporate behavior [39] and reduce their protective behavioral intention. However, other consumers who care more about the facts and technical treatments of the recall will prefer to seek more information to make a decision [39]. Only when they get useful information will they make decisions. Hence, when provided with self-sanction action information, the relationship between consumers’ information seeking and their protective behavioral intention was not clear.

In summary, the disclosure of scientific evidence information may either be regarded as an apology with an explanation [51] and a self-serving behavior [54], or as impressive scientific facts. Therefore, this information will not necessarily lead to information seeking and protective behavioral intention. However, the disclosure of self-sanction action information may overcome the disadvantages of information-sharing actions by directly decreasing consumers’ protective behavioral intention and increasing consumers’ willingness to seek more information. Therefore, the recalling firms can integrate these two types of trust repair strategies to achieve better results.

## 6. Conclusions

This study investigates consumers’ reaction to food recall and estimates the relative effectiveness of two types of trust repair strategies (information-sharing and self-sanction) during a product-harm crisis. In the full model, the results demonstrate that a lower level of food safety trust increases risk perception, CAE, information seeking, and protective behavioral intention. It suggests that the involved firms should take measures to develop consumer food safety trust either in times of crisis or in normal times. For example, the manufacturers and retailers should propose a column in newspapers, on TV, and on professional social media to publicize knowledge of Chinese spirits, such as different Chinese spirit-making technologies and types and functions of legal additives. Furthermore, the firms could post labels on the bottle to show the third-party guarantee (e.g., solid-state fermentation certification). The producers should set up the ‘traceability system of wine quality safety’ to realize the aims of ‘quality safety traceable, risk controllable, product recallable, and liability pursued’, and thus develop consumers’ food safety trust [19]. We also found that consumers exhibiting a high risk and low coping appraisal exhibited a desire for more information, and their information-seeking intention resulted in a higher protective behavioral intention. Hence, managers should take advantage of the information subsidy (i.e., carefully prepare effective information aimed at motivating favorable actions), and distribute this through various channels to influence consumers’ food safety trust, risk perception, CAE, and information seeking [8,29].

Furthermore, we found that consumer trust repair actions and relevant information disclosure are of great importance for recalling companies. Besides traditional trust repair strategies, such as an apology and denial, managers could consider strategies of self-sanction and the sharing of scientific evidence. Managers could release not only self-sanction action messages, such as financial compensation and recall beyond legal liabilities, but also scientific evidence. Given that the self-sanction action messages can repair and develop consumer trust, increasing the tendency for information-seeking, the protective behavioral intention after information-seeking cannot be predicted. Hence, information-sharing actions can remedy the disadvantages of self-sanction behavior only when consumers’ information-seeking requirements are met. Therefore, crisis managers should provide scientific-based information to the public to enhance their knowledge of and trust in the recalled product. However, scientific data supplied by the recalling firm in urgent times may be regarded as self-serving or denial with an explanation, so it would be better for the managers to employ the expert endorsement strategy and ask experts to release information [51]. At the same time, managers should release scientific evidence in normal times, as well as during crisis times.

This study contributes to research on crisis management relating to product recalls. First, this study estimates the consumer crisis-coping process in Chinese spirit recall contexts by integrating the PTM, HSM, and consumer trust theory, which extends the application of PMT from health risk to product harm events. The results support their usefulness in predicting consumer reaction acts in times of a product crisis. Furthermore, this study evaluates the effectiveness of two kinds of information subsidies of trust repair action (self-sanction and information-sharing) based on the theory of consumer trust repair. The results support the hypothesis that consumer responses to product harm crises can be influenced by the trust repair actions adopted by the involved firms.

Some limitations should be acknowledged. First, this study only focuses on product recalls in the food industry, confining the generalization and application of the results. Second, the two measures are provided independently in two scenarios. Further research should examine the relationship between self-sanction and information sharing in a new scenario including these two strategies. Third, the explained variance of PRE is not high, so some other determinants or sub-sections of this construct, i.e., response efficacy and self-efficacy, should be taken into consideration. Furthermore, consumer food safety trust was measured by the three types of trustees, and further research should pay more attention to different types of trustees since previous studies have indicated different types of trust relationships with diversified consumer perceptions of food safety [32]. Therefore, all these matters and other factors should be taken into consideration in future studies.

## Figures and Tables

**Figure 1 ijerph-17-01270-f001:**
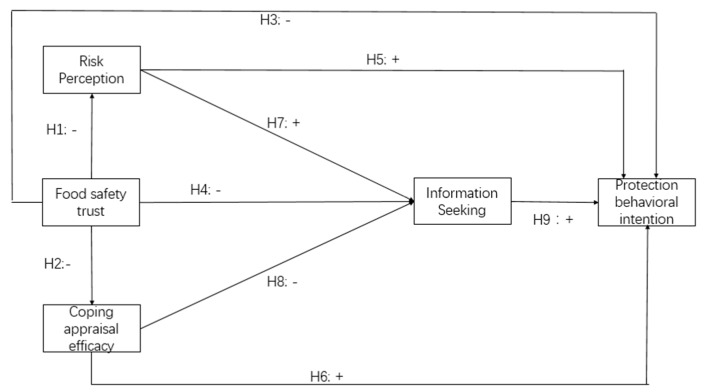
Conceptual model.

**Figure 2 ijerph-17-01270-f002:**
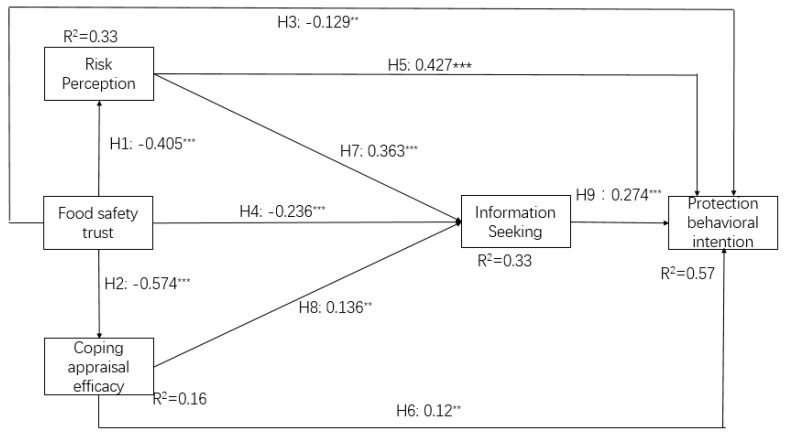
Results of the full analysis. Note: ^**^
*p* < 0.01, and ^***^
*p* < 0.001.

**Figure 3 ijerph-17-01270-f003:**
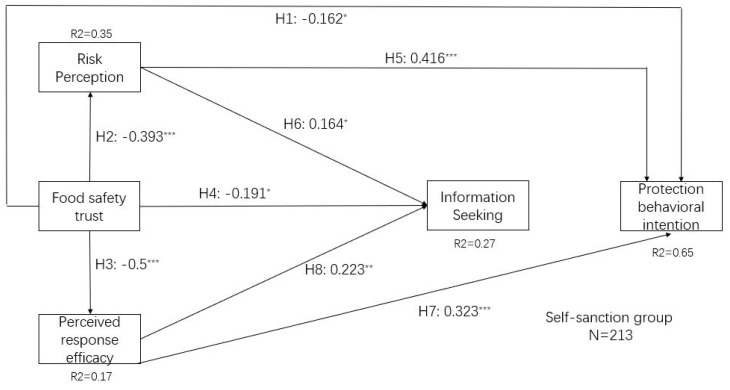
The multiple group analysis results. Notes: χ^2^ = 694.51, df = 300, χ^2^/df = 2.315, *p* < 0.001, TLI = 0.918, NFI = 0.901, CFI = 0.935, RMSEA = 0.054, and RMR = 0.078. Only paths with CR > 2.0 are demonstrated here. * *p* < 0.05, t > 1.96; ** *p* < 0.01, t > 2.58; *** *p* < 0.001, t > 3.29.

**Table 1 ijerph-17-01270-t001:** Constructs, items, and references.

Construct	Item	Measurement	References
Food safety trust (FST)		How much trust do you have in the following institutions or persons that they are conscious of their responsibilities in food safety affairs?	[12]
	FST1	The State Food and Drug Administration and the sub-bureaus.	
	FST2	Food safety experts and scholars.	
	FST3	Food manufacturers and retailers.	
	FST4	Food certification bodies.	
Risk perception (RP)	RP1	It is dangerous to drink the recalled Chinese spirits containing abused additives (such as cyclamate).	[12]
	RP2	Drinking recalled Chinese spirits containing abusive additives, such as cyclamate, would seriously damage my health.	
	RP3	Drinking recalled Chinese spirits containing abusive additives (such as cyclamate) will bring me property loss (possible medical expenses).	
	RP4	When I drink, it is unpleasant to think of Chinese spirits recall originated from abuse of additives, such as cyclamate.	
Coping efficacy perception (CEP)	CEP1	I think change consumption to other Chinese spirits brand can protect my health.	[8,20]
	CEP2	I think stop purchasing the recalled Chinese spirits can save potential medical expenses arising from drinking the recalled Chinese spirits.	
	CEP3	I think I am knowledgeable and competent to handle with the Chinese spirits recall affair.	
	CEP4	I think I have enough time and money to conduct the protective behavior.	
Information seeking (IS)	IS1	I will search for more information about the additive involved in the Chinese spirits recall.	[52]
	IS2	I have to search for more information.	
	IS3	I am concerned with the latest news about the recall every day.	
	IS4	I search for information if it is available in other places.	
Protective behavioral intention (PBI)	PBI1	I would stop purchase the recalled Chinese spirits before I think it is safe.	[9,40]
	PBI2	I would change consumption to other types of wine products (i.e., beers, wines and spirits).	
	PBI3	I will find and consume foreign brand products.	
	PBI4	I will not encourage my acquaintances to buy products from the recalling firms.	

**Table 2 ijerph-17-01270-t002:** Measurement model (confirmatory factor analysis (CFA)) results.

Item	Construct	Estimate	S.E.	C.R.	*p*	R2	Cronbach’s Alpha Value	Composite Liability	AVE
FST1	FST	0.903	0.062	20.906	^***^	0.816	0.93	0.95	0.827
FST2		0.932	0.063	21.51	^***^	0.868			
FST3		0.858	0.047	25.667	^***^	0.737			
FST4		0.761	-	-	a	0.579			
PRE1	PRE	0.945	0.039	27.165	^***^	0.892	0.847	0.897	0.686
PRE2		0.798	0.043	21.707	^***^	0.638			
PRE3		0.921	0.086	10.709	^***^	0.42			
PRE4		0.861	-	-	a	0.742			
RP1	RP	0.77	0.092	13.161	^***^	0.593	0.831	0.888	0.665
RP2		0.77	0.09	13.126	^***^	0.592			
RP3		0.67	0.094	11.876	^***^	0.449			
RP4		0.653	-	-	a	0.426			
IS1	IS	0.713	0.067	14.539	^***^	0.508	0.911	0.937	0.789
IS2		0.791	0.07	16.085	^***^	0.625			
IS3		0.804	0.064	16.235	^***^	0.647			
IS4		0.74	-	-	a	0.548			
PBI1	PBI	0.737	0.059	15.938	^***^	0.543	0.803	0.871	0.629
PBI2		0.842	0.076	14.142	^***^	0.525			
PBI3		0.688	0.06	14.719	^***^	0.474			
PBI4		0.79	-	-	a	0.624			

Note: Regression weight fixed at 1.0. The S.E., C.R., and *p*-value were not estimated in these cases. Extra estimation by fixing a different parameter shown to be statistically significant with *p* < 0.01.

**Table 3 ijerph-17-01270-t003:** Means, standard deviation, and correlations.

Construct	Mean	S.D.	FST	PRE	RP	IS	PBI
FST	3.969	1.022	***0.909***				
PRE	5.257	1.050	−0.582 **	***0.888***			
RP	5.485	0.759	−0.338 **	0.352 **	***0.793***		
IS	4.761	0.855	−0.433 **	0.370 **	0.416 **	***0.828***	
PBI	5.162	0.825	−0.439 **	0.441 **	0.557 **	0.520 **	***0.815***

Note: ^**^
*p* < 0.01. Correlations appear below the diagonal; the diagonal (bold) elements are the square roots of the average variance extracted (AVE), and others are the correlations among constructs.

**Table 4 ijerph-17-01270-t004:** Quadratic regression result, with information seeking being the dependent variable.

Model	Independent	β	t-Value
Model 1	CAE	0.369 ***	8.496
Model 2	CAE	0.843 ***	3.159
	Square of CAE	−0.480 *	−1.986

Note: * *p* < 0.05, and *** *p* < 0.001. CAE: coping appraisal efficacy.

**Table 5 ijerph-17-01270-t005:** Independent sample test between the two groups.

Construct	Information-Sharing Group		Self-Sanction Group		t	Sig.
	Mean	S.D.	Mean	S.D.		
	n = 248		n = 213			
RP	5.552	0.778	5.407	0.73	2.064	0.04
IS	4.864	0.825	4.642	0.877	2.797	0.005
PBI	5.251	0.792	5.058	0.852	2.524	0.012
FST	3.966	0.986	3.972	1.065	−0.26	0.78
PRE	5.274	1.057	5.236	1.045	0.59	0.497

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
