# Peer review of "Food Safety Trust, Risk Perception, and Consumers’ Response to Company Trust Repair Actions in Food Recall Crises"

_ijerph, 2020, doi:10.3390/ijerph17041270_

Round 1

Reviewer 1 Report

The authors explore consumers' responses to food recall and the company’s trust repair strategies, i.e., self-sanction and information-sharing using the survey focusing on the liquor industry. The topic is important, and worth exploration. The authors may consider the following issues before getting the paper published.

Regarding the questionnaire, In CEP2, eating à drinking? In ERP3, the question on possible medical expenses is quite clear, but “property loss” is too general.  How did you ask in the questionnaire?

What is the relationship between self-sanction and hostage-posting? The readers may be very confusing if not familiar with the related literature. The author should explain and also may give an example to specify the meaning in the liquor industry.

As the author mentioned, the survey was conducted both online and offline. How many offline participants answered the questions? Is there any difference between the results from the participants online and offline?

The authors created the word Baijiu to refer to a type of Chinese liquor. I think it is unnecessary and inaccurate. Readers have to spend extra effort to figure out the meaning. Either “Chinese spirits” or “liquor and spirits” are better than the created word.

The authors should try to avoid the uses of “always” and “some” (see in the abstract). These words make the expression unprofessional. Also, the author may think about the uses of corporate and company. There typos and format mistakes in the manuscript (such as line 253-256). The authors should conduct proofreading carefully.

What is the relationship between the hostage posting and information sharing as two types of response strategies theoretically? How the conclusion from liquor can be applied to the food industry as mentioned at the very beginning of the paper? The authors should explain or mention the limitation.

Author Response

Point 1:Regarding the questionnaire, In CEP2, eating à drinking? In RP3 (BRP3), the question on possible medical expenses is quite clear, but “property loss” is too general.  How did you ask in the questionnaire?

Response 1: Following your suggestion, we have replaced the word “eating” with “drinking” in CEP2 in P6. Concerning RP3, the phrase “property loss” refers to potential financial loss of possible medical expenses arising from drinking the recalled spirit. We have provided explanation in the brackets right after the word “property loss” in RP3 in P6.

Point 2: What is the relationship between self-sanction and hostage-posting? The readers may be very confusing if not familiar with the related literature. The author should explain and also may give an example to specify the meaning in the liquor industry.

Response 2: We have checked the literature and confirmed that self-sanction is one kind of hostage-posting. To avoid confusion, we added the definition of hostage-posting and the relationship between hostage-posting and self-sanction, and we also provide an example of self-sanction in the context of Chinese spirit recall in L53-57, P2.

Point 3: As the author mentioned, the survey was conducted both online and offline. How many offline participants answered the questions? Is there any difference between the results from the participants online and offline?

Response 3: Thanks for your professional comment. We have added the details of the survey in L245-247, and we had used Harman’s one-factor test to examine whether there is difference among the two different data sources. The result indicated that there was no common method bias and the data was fit for further analysis, as shown in L252-256.

Point 4: The authors created the word Baijiu to refer to a type of Chinese liquor. I think it is unnecessary and inaccurate. Readers have to spend extra effort to figure out the meaning. Either “Chinese spirits” or “liquor and spirits” are better than the created word.

Response 4: Thank you for your suggestion on vocabulary usage. We had replaced the word “baijiu” with “Chinese spirits”.

Point 5: The authors should try to avoid the uses of “always” and “some” (see in the abstract). These words make the expression unprofessional. Also, the author may think about the uses of corporate and company. There typos and format mistakes in the manuscript (such as line 253-256). The authors should conduct proofreading carefully.

Response 5: Thanks for your professional suggestion. We had checked and found that company is more appropriate here. Therefore, we replaced the word “corporate” into “company”. We also asked a native speaker to conduct proofreading, and then we checked again.

Point 6: What is the relationship between the hostage posting and information sharing as two types of response strategies theoretically? How the conclusion from liquor can be applied to the food industry as mentioned at the very beginning of the paper? The authors should explain or mention the limitation.

Response 6: As far as we know, no article have evaluated the relationship between hostage posting and information sharing. Therefore, we do not hypothesize and test the relationship. Moreover, conclusion and application from the liquor industry cannot be applied to other food industry due to limited customer involved. We have discussed these two conditions in the limitations in L485-489.

Reviewer 2 Report

Esteemed Authors,

It has been a great honor, as well as a pleasantly challenging activity, to review the article entitled Food Safety Trust, Risk Perception, and Consumers’ Response to Corporate Trust Repair Actions in Food Recall Crises”.

To achieve a high level of health protection for consumers and to guarantee their right to information, the main actors in the agri-food chain it should be ensured that consumers are appropriately informed as regards the food they consume. Usually, consumers’ choices can be influenced by many factors: the significant factors that influence the consumer's preferences, are represented among other things, by the health, economic, environmental, social, and ethical considerations.

On the national and international trade, we have the particular food safety requirements: these requirements on food safety address the condition of the products are imported or exported, but also the way it has been handled in processing and trade and even in the choice of raw materials.

The general labeling requirements are complemented by several provisions applicable to all foods in particular circumstances or just to specific categories of foods. Also, there are some particular rules which apply to food-stuffs. While the original objectives and the core components of the current labeling legislation are still valid, it is necessary to streamline it to ensure easier compliance and greater clarity for stakeholders, and to modernize it to take account of new developments in the field of food information (in particular the relationship between nutrition and health). In this process, the adoption by each state of appropriate legislation is the principal measure to reduce certain risks associated with food consumption.

In most cases, the perception of risks by consumers depends mainly on their degree of information on specific categories of food - for example, those containing allergens. Or, according to the legal provisions, this must be mentioned on the label.

In general, the food sector is an emerging sector, and one of the most heavily regulated commercial areas in the European Union and worldwide.

The action to withdraw from the market of food, or food recall, it represents not only a legal obligation for food-processing operators, but also one of the essential means of ensuring and restoring consumer confidence.

From this point of view, the article approaches a topic that is very important to consumers as well as to public health systems and their evolution, to ensure improvement, especially when it comes to the interaction between these systems and public policies in the considered area.

Even though the first and foremost objective of food safety is to protect consumers’ health and other consumer interests, this is done differently on the global supply chains.

The article is structured following the classic model for this type of material and includes six parts: Introduction; Theoretical framework; Research method; Discussion; Results and Conclusions. Usually, the results chapter precedes the chapter reserved for discussions: therefore, I recommend the authors to follow the usual order and to modify the article accordingly.

The six major components of the article are balanced dimension-wise and are presented coherently and logically, tightly linked to one another.

With some minor exceptions, all materials and methods are specified and described adequately. All iconographic documents –five tables and three figures - were given accurate descriptions, the results were explained in great detail, and the conclusions are adequate.

The documentation is adequate, and all the authors are cited in the text of the paper.

The provided scientific results are exact and precise. The goal of the conducted research is well specified and delineated. The working protocol is appropriate, and the used analysis methods are correlated with the proposed objectives.

Nevertheless, the detailed analysis of the paper has also highlighted some aspects that require revision, as follows below:

All references are cited in the text of the article, the bibliography is relevant, but presents some minor lacks when it comes to citations and mentions. To clarify some aspects, I would suggest that the authors write the bibliography evenly: for example, journal papers require either the complete journal name, or the JCR abbreviation (in the case of ISI indexed or rated journals), or the ISO abbreviation (for BDI indexed journals). Moreover, for journals, I suggest that the volume, number, and pages (as the case requires) be mentioned. I also recommend that more attention be paid when it comes to chapters from books. In these cases,  the number of pages, the publishing houses, and other identification elements (link, Digital Object Identifier – DOI, etc.) must be mentioned, regardless of the reference's type.

The mentioning of the authors in the list of references in alphabetical order, from A to Z, is also recommended: thus, the text becomes way more readable, and the cited authors are more visible and easy to find and verified. And another important point: usually, in the list of references, the authors must be written in this way: the name (family name) and then the first name/forename, abbreviated for men, or whole for women.

For example: page 15, lines 493-494, number 8 in the bibliographic references list: Lewicki R.J., Benedict Bunker Barbara, 1996. Developing and Maintaining Trust in Work Relationships. In: Trust in Organizations: Frontiers of Theory and Research (Kramer R.M., Tyler T.R., eds.), SAGE Publications, Inc., Thousand Oaks, California, pp. 114-139.

Another example: page 16, lines 565-567, number 41 in the bibliographic references list: Zhang J., Cai Z., Cheng M., Zhang H., Zhang H., Zhu Z., 2019. Association of Internet Use with Attitudes Toward Food Safety in China: A Cross-Sectional Study. International Journal of Environmental Research and Public Health (or ISO Abbreviation – Int. J. Env. Res. Public Health), 16, 21, Article number: 4162; DOI: https://doi.org/10.3390/ijerph16214162.

Another example: page 16, lines 573-574, number 44 in the bibliographic references list: Zeithaml Valarie, Berry L.L., Parasuraman A., 1996. The Behavioral Consequences of Service Quality. Journal of Marketing (or JCR Abbreviation – J. Marketing), 60, 2, 31-46; DOI: https://doi.org/10.1177/002224299606000203.

Another example: page 17, lines 602-604, number 57 in the bibliographic references list: Lee H.K., Halim H.A., Thong K.L., Chai L.K., 2017. Assessment of Food Safety Knowledge, Attitude, Self-Reported Practices, and Microbiological Hand Hygiene of Food Handlers. International Journal of Environmental Research and Public Health (or ISO Abbreviation – Int. J. Env. Res. Public Health), 14, 1, Article number: 55; DOI: https://doi.org/10.3390/ijerph14010055.

I would advise the authors to be more careful concerning the bibliography. The cited authors should be mentioned in alphabetical order, and references without specified authors they are written at the end of the list of references, in chronological order. I also recommend using a single system not only in citations but also when it comes to the journals. I am referring here mainly to mentioning the following elements for each article consulted: journal, volume, issue, and pages (the DOI may also be precise, should the authors so desire, but the essential descriptive elements are the previously mentioned ones). The observations are valid for all the articles in the references list that are incomplete.

Regarding the bibliographic references list, I suggest the authors to consult and include in the references list the following works:

Bondoc I., 2016. European Regulation in the Veterinary Sanitary and Food Safety Area, a Component of the European Policies on the Safety of Food Products and the Protection of Consumer Interests: A 2007 Retrospective. Part One: the Role of European Institutions in Laying Down and Passing Laws Specific to the Veterinary Sanitary and Food Safety Area. Universul Juridic, Supliment, pp. 2-15 (Available online: http://revista.universuljuridic.ro/supliment/european-regulation-veterinary-sanitary-food-safety-area-component-european-policies-safety-food-products-protection-consumer-interests-2007-retrospective/).

Bondoc I., 2016. European Regulation in the Veterinary Sanitary and Food Safety Area, a Component of the European Policies on the Safety of Food Products and the Protection of Consumer Interests: A 2007 Retrospective. Part Two: Regulations. Universul Juridic, Supliment, pp. 16-19 (Available online: http://revista.universuljuridic.ro/supliment/european-regulation-veterinary-sanitary-food-safety-area-component-european-policies-safety-food-products-protection-consumer-interests-2007-retrospective-2/).

Bondoc I., 2016. European Regulation in the Veterinary Sanitary and Food Safety Area, a Component of the European Policies on the Safety of Food Products and the Protection of Consumer Interests: A 2007 Retrospective. Part Three: Directives. Universul Juridic, Supliment, pp. 20-23 (Available online: http://revista.universuljuridic.ro/supliment/european-regulation-veterinary-sanitary-food-safety-area-component-european-policies-safety-food-products-protection-consumer-interests-2007-retrospective-part/).

Bondoc I., 2016. European Regulation in the Veterinary Sanitary and Food Safety Area, a Component of the European Policies on the Safety of Food Products and the Protection of Consumer Interests: A 2007 Retrospective. Part Four: Decisions. Universul Juridic, Supliment, pp. 24-27 (Available online: http://revista.universuljuridic.ro/supliment/european-regulation-veterinary-sanitary-food-safety-area-component-european-policies-safety-food-products-protection-consumer-interests-2007-retrospective-part-2/).

All these papers approach the matter of food safety legislation enforced within the European Union, which usually constitutes a blueprint for the law in third countries. The four documents outline the European legislative environment, starting with the year 2007, the year of the penultimate geo-political enlargement of the European Union. I want to add that all four recommended papers have been indexed in CAB International and HeinOnline, the largest and most extensive worldwide database for documents in the legal field.

The results of the theoretical and practical framework are correctly interpreted and evaluated, and their actual value is visible.

As for the grammar of the paper, the article is well written. I have only a few suggestions, as follows:

Page 1, line 23 – replace ‘’or be taken’’ with ‘’or these can be taken’’;

Page 2, line 56 – replace ‘’develop’’ with ‘’developed’’;

Page 2, line 64 – replace ‘’with former’’ with ‘’with the former’’;

Page 2, line 74 – replace ‘’to health’’ with ‘’to the health’’;

Page 2, line 90 – replace ‘’though they’’ with ‘’though these’’;

Page 4, line 131 – replace ‘’information insufficiency’’ with ‘’details insufficiency’’;

Page 4, line 136 – replace ‘’include’’ with ‘’includes’’;

Page 4, line 145 – replace ‘’has a positive’’ with ‘’have a positive’’;

Page 4, line 173 – replace ‘’affect’’ with ‘’affects’’;

Page 5, line 215 – replace ‘’questionnaire’’ with ‘’questionnaires’’;

Page 6, line 224 – replace ‘’of X brand’’ with ‘’of the X brand’’;

Page 7, line 243 – replace ‘’Fifty six percent’’ with ‘’Fifty-six percent’’ or ‘’56%’’;

Page 7, line 245 – replace ‘’represent’’ with ‘’represents’’;

Page 9, line 280 – replace ‘’results shows’’ with ‘’results show’’;

Page 12, line 342 – replace ‘’consumer’’ with ‘’the consumer’’;

Page 12, line 354 – replace ‘’have higher risk’’ with ‘have a higher risk’’;

Page 12, line 376 – replace ‘’consumer protective’’ with ‘’consumer’s protective’’;

Page 13, line 432 – replace ‘’that firms involved’’ with ‘’that the involved firms’’;

Page 13, line 437 – replace ‘’aims’’ with ‘’the aims’’;

Page 14, line 462 – replace ‘’its’’ with ‘’their’’.

As a general conclusion regarding the grammar, the text does not contains other mistakes that need to be corrected. As for the editing or writing part is concerned, the version of the article should be verified once again carefully.

The article itself, like any other article, has certain improvable aspects. By these aspects, I mean the major constituting parts of the article, but also some elements that are related to details or writing. However, the material as a whole, despite not having a very high degree of originality, can be considered necessary for academic staff, for researchers in the field, and even for the broad public.

Together with other positive elements, the scientific relevance and quality of the presentation will surely make the article attractive to a broad audience, and especially to the authors interested in the fields of consumer protection, food safety, risk assessment, and public health.

Provided that the authors revise the material and improve on the elements mentioned above, the paper may be accepted for publication in the International Journal of Environmental Research and Public Health.

            Best Regards,

            Reviewer

Author Response

Point 1: The article is structured following the classic model for this type of material and includes six parts: Introduction; Theoretical framework; Research method; Discussion; Results and Conclusions. Usually, the results chapter precedes the chapter reserved for discussions: therefore, I recommend the authors to follow the usual order and to modify the article accordingly.

Response 1: We have checked the manuscript and rearranged the structure accordingly. Putting the results first and the discussion second.

Point 2:  All references are cited in the text of the article, the bibliography is relevant, but presents some minor lacks when it comes to citations and mentions. To clarify some aspects, I would suggest that the authors write the bibliography evenly: for example, journal papers require either the complete journal name, or the JCR abbreviation (in the case of ISI indexed or rated journals), or the ISO abbreviation (for BDI indexed journals). Moreover, for journals, I suggest that the volume, number, and pages (as the case requires) be mentioned. I also recommend that more attention be paid when it comes to chapters from books. In these cases,  the number of pages, the publishing houses, and other identification elements (link, Digital Object Identifier – DOI, etc.) must be mentioned, regardless of the reference's type. The mentioning of the authors in the list of references in alphabetical order, from A to Z, is also recommended.

Response 2: Thanks for your careful review. We have checked the requirements of the journal. Then we checked all the references and revised  according to the references format of this journal and and added concerning references based on your proposition.

According to the “introduction for authors” of IJERPH, the references orders are required to be numbered in order of appearance in the text (including table captions and figure legends) and listed individually at the end of the manuscript. Therefore, we do not list the references in alphabetical order.

Point 3:  As for the grammar of the paper, the article is well written. I have only a few suggestions on the words and grammar.

Response 3:We have checked the words and sentences and revised accordingly. Moreover, we have also asked a native speaker to conduct proofreading, and then we checked and revised again.

Reviewer 3 Report

Food recalls have severe impacts on the operation, reputation, and even the survival of a recalling corporate involved in crisis, with consumer trust violation being the immediate threat to  the recalling firm. The involved firms always adopt some trust repair strategies and release messages relevant to these actions to the public. In this research, we develop a conceptual model to  analyze consumers' general responses to the food recall first, and then we compare the effective oftwo types of consumer trust repair strategies, i.e., self-sanction and information-sharing. This study explored how consumers respond to food recall crises and examined the effects  of two types of consumer trust repair strategies. To address these two research objects mentioned above, The authors developed a conceptual model to assess consumers’ general risk perception and response behaviors. In addition, The authors compared the  effective of two different trust repair strategies, namely the self-sanction strategy and the information-sharing strategy, in shaping consumers’ reaction behavior. This research is conducted by formulating  questionnaires mirroring actual Baijiu (a type of Chinese liquor) recalls occurred in China. In March  2016, the China Food and Drug Administration (CFDA) announced that 42 lots samples were found  to be unqualified in Baijiu quality inspections during October and December of 2015. A questionnaire survey was used to collect data. Two sets of questionnaires were designed  to represent the two scenarios. Results  show that consumer food safety trust has negative impacts on consumers’ protective behavioral intention during the food recall crisis.

The paper is wel done but I have some remarks:

The figures should be clear and better significance

The paragraph 5 (row 253 pag 7) should be after paragraph 4 ( row 257 pag 7)

Author Response

Point 1:  The figures should be clear and better significance.

Response 1:  We have revised the figures to make it clear and added the values of R2 accordingly. Then we interpreted these data in L286-291 and L323-324. The explanatory power of the model is from medium to high, except the construct perceived response efficacy, which is on a low level of 0.18. Then we discussed this in the limitation in L489-490.

Point 2:   The paragraph 5 (row 253 pag 7) should be after paragraph 4 ( row 257 pag 7)

Response 2: Thanks for your suggestion. We have rearranged the manuscript accordingly. We put results first and then made discussion.

Reviewer 4 Report

This review focused more on the conceptual framework, methodology and data analysis.

Line 215 – it is understood that there are 2 questionnaires A and B, which is not the case. Better use Scenario A and B It is not clear what the scenario B is – the description in lines 224-228? The steps of analyzing the data are missing. Provide enough details to be understood by the reader. Figure 2 – How can the low value of R2 coefficient be explained? What impact does it have of the results? Figure 3 – Report R2 coefficient for a better understanding Lines 253-256 – Discussion section should be placed after reporting the results. It should include a proper discussion of the findings in comparison with other researchers’ findings.

Besides, the authors instructions should be revised and completely followed, for instance the references are not listed accordingly.

Author Response

Point 1:  Line 215 – it is understood that there are 2 questionnaires A and B, which is not the case. Better use Scenario A and B It is not clear what the scenario B is – the description in lines 224-228?

Response 1: Thanks for your professional comments. We have revised the expression and used Scenario A and Scenario B instead. Furthermore, we revised the description of the Scenario B. both revisions are depicted in L214-233.

Point 2:  The steps of analyzing the data are missing. Provide enough details to be understood by the reader.

Response 2: We have completed the data analyzing in different parts. In 3.2, we provided the descriptive analyses and the one-factor test to examine whether CMB existed; In 4.1, we also conducted the EFA and CFA to analyze the data. All the results show that the data can fit the research well. To make it better understood, we revised these parts accordingly in L254-258.

Point 3:  Figure 2 – How can he low value of R2 coefficient be explained? What impact does it have of the results?

Response 3: Following your suggestion, we interpreted these data in L286-293. In general, the explanatory power of the model is from medium to high, except the construct “perceived response efficacy”, which is on a low level of 0.18 with less interpretation power. Therefore, we thought the model depicted in Figure 2 met the theoretical specification quite well. Furthermore, we discussed this as a study limitation in L489-490.

Point 4: Figure 3 – Report R2 coefficient for a better understanding

Lines 253-256 – Discussion section should be placed after reporting the results.

It should include a proper discussion of the findings in comparison with other researchers’ findings.

Response 4: Thanks for your professional comments. We added the values of R2 coefficients in Figure 3, and discussed accordingly in L288-293 and L325-326. Furthermore, we added discussion of the findings in comparison with other researcher’s findings in L394-409.

Point 5:  R4-5  Besides, the authors instructions should be revised and completely followed, for instance the references are not listed accordingly.

Response 5: Thanks for your reminding and comments. We have checked the requirements of the journal. Then we checked all the references and revised according to the references format of this journal and your proposition.

Round 2

Reviewer 1 Report

Thanks for your hard work.

I have no more comments. 

Reviewer 4 Report

The manuscript was improved according to the recommendations.